# Psychopathy, pain, and pain empathy: A psychophysiological study

**Sophie Alshukri**[1]*, **Minna Lyons**[1], **Victoria Blinkhorn**[1], **Luna Muñoz**[2☯], **Nicholas Fallon**[3☯]

1 School of Psychology, Faculty of Health, Liverpool John Moores University, Liverpool, United Kingdom,
2 The Luminary Group Ltd., Liverpool, United Kingdom, 3 Department of Psychology, Institute of Population
Health, University of Liverpool, Liverpool, United Kingdom

☯ These authors contributed equally to this work.
* s.alshukri@2022.ljmu.ac.uk

pone.0306461

CANADA

**Data Availability Statement:** The data underlying
the results presented in the study have been
uploaded in the "attach files" section. Data can be
found as supporting information files.

## Abstract

The present study examined whether people higher in psychopathy experienced less self-
reported and psychophysiological nociceptive pressure than people lower in psychopathy.
We also examined whether psychopathy affects empathy for others' pain via self-reported
and psychophysiological measures. Three hundred and sixty-nine students (18–78 years;
$M = 26$, $SD = 9.34$) were screened for psychopathic traits using the Youth Psychopathy
Inventory (YPI). Stratified sampling was used to recruit 49 adults residing in the highest ($n = $
23) and lowest ($n = 26$) 20% of the psychopathy spectrum. Using skin conductance
response (SCR) and self-report responses, participants responded to individually adjusted
intensities of pneumatic pressure and others' pain images and completed self-reported psy-
chopathy and empathy measures (Triarchic Psychopathy Measure, TriPm; Interpersonal
Reactivity Index, IRI). People higher in psychopathy self-reported feeling less nociceptive
pressure compared to people lower in psychopathy, yet we did not find any differences in
SCR to nociceptive pressure. However, when viewing other people in pain, the high psy-
chopathy group displayed lower SCR and lower self-reported empathy compared to those
lower in psychopathy. Our results suggest psychopathic traits relate to problems empathis-
ing with others' pain, as well as the perception of nociceptive pressure. We also show sup-
port for the theory of dual harm which has been receiving increasing attention.
Consequently, psychopathy interventions should focus both on recognising and empathis-
ing with the pain of others.

## Introduction

Psychopathy is a personality trait that has been related to multiple adverse outcomes, including
aggression towards others [e.g. 1] as well as aggression towards oneself [2]. The triarchic
model of psychopathy divides it into three factors: boldness (i.e. social dominance, emotional
resiliency), meanness (i.e. low empathy, exploitativeness), and disinhibition [i.e. low impulse
control; 3]. People at the higher end of the psychopathy spectrum typically have trouble recog-
nising their own emotions as well as the emotions of others [4]. Indeed, it is possible that the
inability to recognise one's own emotions stems from a poor recognition of others' emotions,

**Funding:** The authors received no specific funding for this work.

**Competing interests:** The authors have declared no competing interests exist.

contributing to low empathy [5]. Interestingly, psychopathy (especially meanness) has also been associated with low empathy for the pain of others [6] as well as increased nociceptive pain tolerance [7, 8]. As a result, the present study aimed to further the knowledge of a link between psychopathy, experiencing nociceptive pressure, and empathy for others' pain.

There are several reasons to investigate pain in the context of psychopathy. Firstly, pain and distress are typically communicated through facial expressions and vocalisations to attract help [9]. However, psychopathy relates to a reduced ability to recognise distress and pain in others, as well as lower levels of prosocial behaviours needed to help those individuals [10, 11]. Yet, when psychopathic offenders were asked to empathise, they showed relatively normal levels of empathy [12]. In addition, findings indicate deficits in brain regions associated with processing distress cues in individuals with psychopathy, which may impact empathic responses to others' pain [13]. Furthermore, ones' own pain distress was found to influence views about how much pain another person experienced, with higher scores on lifestyle (or disinhibition) facet of psychopathy predicting lower estimates of other's distress levels [14]. Since recognising others in pain and prosocial responses are related, it's important to investigate psychopathy and empathy for others' pain.

Second, psychopathy shares co-morbidity and risk factors with both self-harm and aggression towards others [15]. According to the dual harm model, the co-occurrence of self-harm and aggression could relate to emotional dysregulation [15], which could also link to diminished perception of pain [16]. This indicates that psychopathy could relate to both reduced empathy for the pain of others, as well as perception of pain for the self.

Third, observing and experiencing nociceptive pain relies on affective empathy [17], which could have neural bases in the mirror neuron system [18]. Indeed, research has shown that similar neural networks are activated when observing others in pain and when experiencing nociceptive pain in typically developing individuals [see 19]. Psychopathy is associated with both low affective empathy [e.g. 20, but also see 21], as well as hypoactivation of the mirror neuron system when observing others in pain [18]. These findings indicate that psychopathy may influence pain perception and pain empathy for the self, as well as for other people.

Fourth, current literature suggests mixed findings on psychopathy, nociceptive pain, and pain empathy for others depending on the methods used (e.g. self-report vs behavioural measures). For instance, some studies have found that aspects of psychopathy relate to increased tolerance of pressure stimuli and electric shocks [8, 22], whereas others have not found this [23]. Psychopathy was also correlated with increased nociceptive pain tolerance in studies using self-report measures [23, 24]. In addition, psychopathy has been associated with blunted neural responses to the pain of others [25, 26], but not when imagining pain in the self [26]. Yet, one study found that although psychopathy had a link with a decreased ability to assess pain expressions, it did not relate to self-reported pain attributions to others [27]. These results suggest individuals with psychopathy process pain in the self differently than the pain of others. As such, by utilising both an objective measure (skin conductance response, or SCR) and a self-reported measure, the current study can look at the differences when administering nociceptive stimuli and empathy images in psychopathy.

SCR (an indirect measure of sympathetic nervous activity) can measure emotional arousal which may be related to nociceptive pain experience [28, 29]. Research has found psychopathy and callous-unemotional traits [the affective dimension of psychopathy; 30] are associated with lower SCR to fear-inducing stimuli, suggesting diminished responses to threat [31–33]. Moreover, violent incarcerated offenders had reduced SCR when viewing others in pain [34]. These findings stress the importance of assessing psychophysiology in psychopathy. As such, individuals high in psychopathy may be unable to respond to emotionally salient and arousing stimuli, therefore, not respond to others' emotions, including experiencing nociceptive pain.

In addition, psychopathy could be related to a discrepancy in physiological and self-reported responses to directly experienced nociceptive pain and the pain of others. It has been suggested that psychopathy is linked to somatic aphasia; the inaccuracy in identifying and recognising somatic states of the self [35]. Indeed, research in incarcerated men [34] and children at high-risk of criminal behaviour [36] suggests that psychopathy is associated with blunted physiological, but not self-reported responses of empathy to others' pain. To the best of our knowledge, there are no studies using adult community samples that have simultaneously looked at both self-reported and psychophysiological responses to directly experienced nociceptive pressure stimuli, and pain empathy for other people. Based on somatic aphasia, it would be expected that individuals with high levels of psychopathy may differ in their responses to nociceptive pressure and the pain of others when arousal is measured with SCR, and self-reported measures.

The present study sought to investigate psychopathy and its relationship to self-report measures and SCR to directly experienced nociceptive pressure stimuli, and how it relates to recognising the pain of others. Although psychopathy and empathy are well studied, and deficits in both self-report measures and physiology are seen [e.g. 25, 34], pain perception is not fully explored, with little research investigating physiological aspects. By understanding this aspect more, and incorporating the dual harm model [15], we may understand more about why and how individuals with psychopathy show a reduction in empathy. The following research questions were addressed:

1. Do people higher in psychopathy experience less intense nociceptive pain to pressure stimuli than people lower in psychopathy via self-report responses and SCR?

2. Do people higher in psychopathy feel less empathy for other people's pain via self-report responses and SCR?

## Methods

### Participants

Three-hundred and sixty-nine students (18–87 years; $M = 26$, $SD = 9.34$) were recruited between June 2018 and March 2019 via advertisements located around the University of Liverpool campus such as on notice boards and in communal areas. Those interested were screened for psychopathic traits using an online version of the Youth Psychopathic Inventory [YPI; 37]. A stratified sampling technique was used to invite potential participants who scored in the highest and lowest 20% of the psychopathy spectrum to a research study in the laboratory. One hundred and thirty-one participants (low psychopathy $n = 63$; YPI $M = 73$, $SD = 6.08$; high psychopathy $n = 68$; YPI $M = 132$, $SD = 10.05$) were contacted to take part in the laboratory experiment; a total of 49 adults (female $n = 26$; male $n = 23$), aged between 18–55 years old ($M = 25$, $SD = 7.03$) accepted the invitation to take part (low psychopathy $n = 26$, min YPI score = 71, max YPI score = 90, $M = 80$, $SD = 5.02$; high psychopathy $n = 23$, min YPI score = 120, max YPI score = 149, $M = 133$, $SD = 8.13$). Participants' data were anonymised by assigning a number to each dataset and keeping all identifying paperwork in a locked storage space that only the supervising investigator (NF) had access to.

### Procedure

Ethical approval was granted by the University of Liverpool's Ethics Committee. Participants invited into the laboratory were seated in a chair positioned approximately 80cm from a 48.2cm (19-inch) Dell OptiPlex 780 computer monitor (S1 Fig). Following consent,

participants were fitted with electrodes to measure SCR. An individual mould was made of dental putty to ensure the finger remained consistent throughout the task. Prior to the experiment, participants were given 2 self-report questionnaires [IRI and TriPm; 38, 39] to complete to accommodate a 10-minute stabilisation period for SCR.

Participants were given a demonstration of the pressure stimulator which created nociceptive pressure before the experimental program began. For the experiment, participants positioned the index finger of their dominant hand in the mould under the circular probe while they rested their non-dominant hand on the table. The probe covered the lunular of the fingernail and adjacent skin and was lowered onto this area to create pressure [40]. Participants received training to select an individualised appropriate level of pressure to evoke a moderate self-reported pain response for the task [adapted staircase procedure; 41]. The intensity of each pressure stimulus, measured in volts (v), was rated on a 0–100 numerical rating scale (NRS; 0 representing no pain or sensation at all, 100 representing the most pain imaginable). This was verbally explained and presented in visual form [self-assessment manikin or SAM; 42]. The pressure level was gradually increased in small increments (0.1–0.2 v) until pain threshold (3 on pain scale) and moderate pain (6/7 on pain scale) was reached for each participant.

Participants experienced 10 touch stimulations (also referred to as no pressure since it was a touch sensation, attained by calculating 1/3 of moderate pressure level), 10 threshold pressure stimulations, and 10 moderate pressure (also referred to as high pressure) stimulations in a pseudorandom order. Each trial began with a 3-second rest interval period where participants viewed a white fixation cross on a grey background, followed by a grey screen which signalled pressure stimulation. Full pressure lasted for 1 second, followed by an immediate release [40]. Participants then rated physical pressure intensity ranging from "no pain/sensation" (0) to "worst pain imaginable" (100) using a NRS on the screen. Participants were instructed to keep their finger in the mould until the NRS appeared, remove it to rate their self-reported pain, then place it back in the mould. A grey screen would appear to prompt participants to place their finger under the probe and prepare for the next stimulation. Participants were made aware of the safety features of the stimulator and could abort the process at any time by removing their finger from the machine. The task lasted approximately 15 minutes.

The empathy task was similar to previous studies [see 43, 44]. Each trial began with a white fixation cross on a grey background. Participants had a single viewing of 30 images. The images, originally developed for Fallon et al. [45], consisted of fifteen pictures containing feet or hands depicting painful situations, such as a hand trapped in a car door, or a foot standing on a fractured piece of glass, and 15 images depicting non-painful scenes graphically matched but contained no pain, for example, a hand next to a car door rather than trapped in it, and a foot placed safely on the ground with no signs of broken debris. Each image was presented for 5 seconds. After each image, a 5 second computerised response period followed. Participants were asked to rate how much pain they perceived using a NRS ranging from "no pain" (0) to "worst possible pain" (100). The images were presented in a pseudo-randomised order, and the task lasted approximately 10 minutes. Cronbach's alpha ($\alpha$) coefficient for self-report responses to non-pain ($\alpha$ = .79) were rendered acceptable, whereas self-report responses to pain images ($\alpha$ = .96) had excellent internal consistency.

Participant data such as demographic information, questionnaire responses and SCRs were anonymised by assigning a number to each dataset making it unidentifiable, and all identifying paperwork was kept in a locked storage space that only the supervising investigator (NF) had access to.

**Skin conductance response.** To measure skin conductance, two Ag-AgCl electrodermal conductance electrodes containing 0.5% chloride gel concentration were attached to the volar surface of the index and middle distal phalanges (finger pads of the index and middle finger) for the most reliable electrodermal activity measurement [28], and secured with surgical tape. Data were recorded using a MindWare Mobile Impedance device (Mindware Technologies Ltd., Gahanna, Ohio, USA). The device transmitted physiological signals wirelessly and remotely via Billion BiPAC 5200G router (Billion Electric Co., Ltd., London) to a HP Notebook laptop running Biolab Acquisition software (Mindware Technologies Ltd.). SCR was recorded using a low-pass filter of 1 hertz (Hz) and a gain of 5 μS/V. The waveform was smoothed at 500 samples. Data were analysed offline using Mindware Technologies' Electrodermal Activity (EDA) analysis software application. Event-related SCR was used to identify discrete responses following a pressure/pain (or non-pressure/pain) event. SCR for the self-reported pain to pressure task was calculated by identifying the peak of skin conductance within the latency window of 1 to 4 seconds after the release of the pressure probe (S2 Fig). SCR for the empathy task was calculated in a similar way but by identifying the peak of skin conductance within the latency window of 1 to 4 seconds after the presentation of the image (S3 Fig). There was no missing data in the sample. Cronbach's alpha coefficient for SCR to low ($\alpha = .62$) and high ($\alpha = .17$) pressure showed acceptable and poor internal consistency respectively, whereas SCR to pain ($\alpha = .66$) and non-pain images ($\alpha = .63$) were rendered acceptable.

**Pneumatic pressure stimulator.** Nociceptive pressure was delivered using a pneumatic pressure stimulator designed by Dancer Design (St. Helens, UK). The system included a pneumatic force controller which used compressed air from a 11.1 litre aluminium cylinder to lower a 1 cm$^2$ circular probe with variable force. Each stimulus was delivered by passing a specific voltage into the pressure stimulator, which translates into pressure in a range from 0.00 kg/cm$^2$ (generated from 0.00 v input) to a maximum of 3.5 bar (11.55 kg/cm$^2$, generated from 3.5 v input) to avoid injury. Voltages were generated by a computer program written in PsychoPy in Python programming language (LabJack Corp., Lakewood, CO, USA). Cronbach's alpha coefficient for self-report responses to touch/no pressure ($\alpha = .91$) and high ($\alpha = .93$) pressure showed excellent internal consistency.

**Psychopathic traits.** The Youth Psychopathic Inventory [YPI; 37] was used to screen participants for psychopathy traits. The YPI is a 50-item self-report measure designed to assess 10-core concepts related to psychopathy, each containing 5 items; dishonest charm (e.g. "It's easy for me to charm and seduce others to get what I want from them"; $\alpha = .90$); grandiosity (e.g. "I'm better than everyone on almost everything"; $\alpha = .85$); lying (e.g. "Sometimes I find myself lying without any particular reason"; $\alpha = .89$); manipulation (e.g. "I can make people believe almost anything"; $\alpha = .93$); remorselessness (e.g. "I seldom regret things I do, even if other people feel that they are wrong"; $\alpha = .90$); unemotionality (e.g. "what usually scares others usually doesn't scare me"; $\alpha = .78$); callousness (e.g. "I think that crying is a sign of weakness, even if no one sees you"; $\alpha = .80$); thrill-seeking (e.g. "I like to be where exciting things happen"; $\alpha = .75$); impulsiveness (e.g. "I prefer to spend my money right away rather than save it"; $\alpha = .76$) and irresponsibility (e.g. "I have often been late to work or classes in school"; $\alpha = .75$). Items were scored on a 4-point Likert scale from "does not apply at all" (1) to "applies very well" (4). Cronbach's alpha score for the YPI ($\alpha = .95$) and its subscales showed adequate reliability similar to previous research [31, 46].

The Triarchic Psychopathy Measure [TriPm; 38] was used to assess and confirm psychopathy scores once in the laboratory. The TriPm is a 58-item self-report measure designed to assess psychopathy using three distinct constructs; boldness (e.g. "I am well-equipped to deal with stress"), meanness (e.g. "I enjoy a good physical fight"), and disinhibition (e.g. "I jump into things without thinking"). Items are scored on a 4-point Likert scale from "true" (3) to

"false" (0). Cronbach's alpha coefficient was strong for TriPm total score ($\alpha$ = .94) as well as for each of the constructs ($\alpha$ = .88, $\alpha$ = .94, $\alpha$ = .87, respectively).

**Empathic traits.** The Interpersonal Reactivity Index [IRI; 39] was used to assess self-reported empathy. The 28-item self-report measure assesses empathy using 4 subscales; perspective taking (e.g. "I try to look at everybody's side of a disagreement before I make a decision"), fantasy (e.g. "I really get involved with the feelings of the characters in a novel"), empathic concern (e.g. "I am often quite touched by things that I see happen"), and personal distress (e.g. "In emergency situations, I feel apprehensive and ill-at-ease"). Items are scored on a 5-point Likert scale from "does not describe me well" (0) to "describes me very well" (4). The self-report measure yielded a good internal consistency using Cronbach's alpha coefficient ($\alpha$ = .85) overall as well as for each of the constructs ($\alpha$ = .76, $\alpha$ = .71, $\alpha$ = .85, $\alpha$ = .80, respectively). Fantasy was not used for data analysis as we were not looking at participants' ability to adopt the thoughts and feelings of fictitious characters from books, movies or plays.

## Data analysis plan

To test whether people higher in psychopathy experienced less intense nociceptive pain than people lower in psychopathy when given individually adjusted pressure intensities to report the same subjective pain intensity (i.e. moderate pain), we performed a 2-way mixed ANOVA with pressure intensity (touch/no pressure, high pressure) as a repeated measures factor (dependent variable; DV) and psychopathy group (low, high) as a between subject's factor (independent variable; IV). This was performed for both NRS self-report and SCR data. To test whether people higher in psychopathy felt less empathy for other people's pain, we again performed a 2-way mixed ANOVA with empathy images (no pain, pain) as a repeated measures factor (DV) and psychopathy group (low, high) as a between subject's factor (IV). This was again performed for both NRS self-report and SCR data.

To test whether people higher in psychopathy required objectively more intense pressure stimuli (measured in volts) to report the same subjective pain intensity (i.e. moderate pain) as the low psychopathy group, we performed an independent sample's t-test with psychopathy group (low, high) as the IV and pressure stimuli level (moderate pressure) as the DV. We used Hedge's g correction for effect sizes and 95% confidence intervals (CI) as it uses a correction factor for small sample sizes [47].

We also ran manipulation checks to ensure the effectiveness of our study. To test whether our high psychopathy group scored significantly higher on psychopathy facets and lower on empathy facets compared to our low psychopathy group, we ran an independent sample's t-test using Hedge's g correction. We used psychopathy group (low, high) as IV, and the subscales of the TriPm and IRI as DVs. Analysis was conducted in JASP (version 0.18.3, JASP Team, 2024).

## Results

### Tests of normality

The distribution of SCR to touch/no pressure ($Z$ = 5.79) and high nociceptive pressure ($Z$ = 1.94) as well as observing others' pain images ($Z$ = 3.18) and observing other's non-pain images ($Z$ = 4.00) were negatively skewed. Due to this, a square root transformation was conducted to ensure that the data follow approximately normal distribution for each of the SCR variables; touch/no pressure (skewness = 1.75, $SE$ = .340, Z-skewness = 5.15), high pressure (skewness = .573, $SE$ = .340, Z-skewness = 1.69), observing others' pain images (skewness = -.611, $SE$ = .340, Z-skewness = -1.80), observing other's non-pain images (skewness = 1.28, $SE$ = .340, Z-skewness = 3.76).

### Main study questions

**Do people higher in psychopathy experience less intense nociceptive pain to pressure stimuli than people lower in psychopathy?.** We examined whether people higher on psychopathy would report less intense subjective (NRS) nociceptive pain when given individually adjusted pressure intensities to report the same subjective pain intensity (i.e. moderate pain), and whether that would also be reflected in their SCR. For NRS self-report data, the repeated measures ANOVA showed a significant effect of pressure intensity, $F(1, 45) = 228.54$, $p < .001$, $\eta_p^2 = .84$, a significant between-subjects effect of psychopathy group, $F(1, 45) = 7.58$, $p = .008$, $\eta_p^2 = .144$, and a non-significant interaction effect between pressure intensity and psychopathy group, $F(1, 45) = .71$, $p = .40$, $\eta_p^2 = .02$. Post hoc tests showed high levels of pressure were rated as significantly higher ($M = 58.20$, $SE = 2.00$) than lower levels of pressure ($M = 16.1$, $SE = 1.61$), $t(45) = 2.79$, $p = .008$, 95% CI [2.55, 4.23], Cohen's $d = 3.40$. In addition, the high psychopathy group reported experiencing significantly less pain ($M = 33.96$, $SD = 1.70$) compared to the low psychopathy group ($M = 40.40$, $SD = 1.60$), $t(47) = 2.75$, $p < .01$, 95% CI [.12, .91], Cohen's $d = .52$.

For SCR, there was a significant effect of pressure intensity, $F(1, 47) = 4.45$, $p = .04$, $\eta_p^2 = .09$, a non-significant between subjects effect of psychopathy group, $F(1, 47) = .22$, $p = .64$, $\eta_p^2 = .005$, and a non-significant interaction effect between pressure intensity and psychopathy group, $F(1, 47) = 3.04$, $p = .09$, $\eta_p^2 = .06$. Post hoc comparisons showed high levels of nociceptive pressure produced greater SCR ($M = 1.07$, $SE = .01$) compared to lower levels of nociceptive pressure ($M = 1.05$, $SE = .01$), $t(47) = 2.11$, $p = .04$, 95% CI [.01, .62], Cohen's $d = .31$.

We tested whether people higher in psychopathy ($M = 2.61$, $SD = .74$) required objectively more intense pressure stimuli (measured in volts) to report the same subjective pain intensity (moderate pain) as the low psychopathy group ($M = 2.30$, $SD = .52$), however this was non-significant, $t(47) = -1.70$, $p = .1$, 95% CI [-1.05, .09], Hedge's $g = -.48$.

**Do people higher in psychopathy feel less empathy for other people's pain?.** Using the images of other's experiencing pain, we tested if those higher in psychopathy would rate the images as less painful than those lower in psychopathy. We also tested whether SCR would be lower for those higher in psychopathy when viewing images of others' pain. The repeated measures ANOVA showed a significant effect of pain intensity (pain images, non-pain images), $F(1, 47) = 188.48$, $p < .001$, $\eta_p^2 = .56$, a significant between-subjects effect of psychopathy group, $F(1, 47) = 10.21$, $p = .002$, $\eta_p^2 = .18$, and a significant interaction effect between pain intensity and psychopathy group, $F(1, 47) = 12.73$, $p < .001$, $\eta_p^2 = .04$. Post hoc tests showed overall, pain images were rated as more painful ($M = 45.60$, $SE = 3.14$) compared to non-pain images ($M = 5.61$, $SE = .91$), $t(47) = -.13.7$, $p < .001$, 95% CI [1.85, 3.04], Cohen's $d = 2.47$. In addition, the low psychopathy group had more empathy for other's pain images ($M = 31.4$, $SE = 2.48$) compared to the high psychopathy group ($M = 19.8$, $SE = 2.63$), $t(47) = 3.20$, $p = .002$, 95% CI [.24, 1.20], Cohen's $d = .71$. Lastly, the interaction effect showed that the high psychopathy group self-reported less empathy to pain images ($M = 34.61$, $SE = 4.57$) compared to the low psychopathy group ($M = 56.56$, $SE = 4.30$), $t(47) = 4.64$, $p < .001$, 95% CI = [.49, 2.21], Cohen's $d = 1.35$. However, the high psychopathy group did not self-report significantly less empathy to non-pain images ($M = 5.03$, $SE = 1.41$) compared to the low psychopathy group ($M = 6.19$, $SE = 1.33$), $t(47) = .25$, $p < .060$, 95% CI [-.70, .85], Cohen's $d = .07$.

For SCR, there was a significant effect of pain intensity (pain images, non-pain images), $F(1, 47) = 453.63$, $p < .001$, $\eta_p^2 = .91$, a significant between-subjects effect of psychopathy group, $F(1, 47) = 12.83$, $p < .001$, $\eta_p^2 = .21$, and a significant interaction effect between pain intensity and psychopathy group, $F(1, 47) = 13.13$, $p < .001$, $\eta_p^2 = .22$. Post hoc tests showed that pain images produced greater SCR ($M = 6.53$, $SE = .26$) than non-pain images ($M = 1.04$, $SE = .01$),

**Table 1. Independent sample's t-tests for psychopathy and empathy subscales.**

| In-lab measures | Low | | High | | $t(47)$ | $p$ | 95% CI | | Hedge's g |
|---|---|---|---|---|---|---|---|---|---|
| | *M* | *SD* | *M* | *SD* | | | Lower | Upper | |
| Boldness | 24.9 | 8.16 | 36.3 | 8.67 | -4.75 | < .001 | -2.00 | -.71 | -1.34 |
| Meanness | 13.9 | 13.97 | 28.0 | 7.86 | -4.28 | < .001 | -1.81 | -.60 | -1.21 |
| Disinhibition | 13.5 | 10.87 | 21.3 | 8.02 | -2.83 | .007 | -1.40 | -.21 | -.80 |
| Empathic Concern | 22.0 | 3.18 | 15.8 | 4.47 | 5.59 | < .001 | .92 | 2.21 | 1.60 |
| Perspective Taking | 20.2 | 4.60 | 17.5 | 4.82 | 2.04 | .047 | -2.06 | 1.15 | .58 |
| Personal Distress | 15.3 | 5.46 | 11.6 | 5.53 | 2.40 | .021 | .10 | 1.25 | .68 |

The table shows low psychopathy and high psychopathy groups' respective mean (*M*) and standard deviation (*SD*), t-statistic and degrees of freedom t(*df*), p value (*p*), lower and upper 95% confidence intervals (95% CI), and Hedge's *g* of effect size for each subscale.

$t$ (47) = -21.3, $p$ < .001, 95% CI [3.35, 5.30] Cohen's $d$ = 4.32. Overall, the low psychopathy group produced greater SCRs (*M* = 4.24, *SE* = .18) compared to the high psychopathy group (*M* = 6.53, *SE* = .26), $t$ (47) = -21.3, $p$ < .001, 95% CI [3.35, 5.30] Cohen's $d$ = 4.32. Lastly, the high psychopathy group produced significantly lower SCRs to pain images (M = 5.60, SD = .37) compared to the low psychopathy group (*M* = 7.45, *SE* = .35), $t$ (47) = 5.10, $p$ < .001, 95% CI [.59, 2.33], Cohen's $d$ = 1.46. However, non-pain images did not produce a significant effect between low (*M* = 1.03, *SE* = .01) and high psychopathy groups (*M* = 1.05, *SE* = .01), $t$ (47) = -.05, $p$ = .170, 95% CI [-.80, .76], Cohen's $d$ = -.01.

**Manipulation checks.** We tested if our high psychopathy group scored higher in psychopathy than our low psychopathy group using a separate psychopathy measure [TriPm; 38]. We found the high psychopathy group scored significantly higher on all subscales of boldness, meanness, and disinhibition when compared to the low psychopathy group. The high psychopathy group also reported significantly lower empathic concern, personal distress, and perspective taking when compared to the low psychopathy group (Table 1).

## Discussion

Our study aimed to investigate psychopathic traits, their relationship to SCR and NRS self-reported responses to directly experienced pressure in oneself, and how it may relate to empathising with the pain of others. We found people in the high psychopathy group had less self-reported empathy and lower SCR for other people's pain. Additionally, we found people in the high psychopathy group self-reported experiencing less intense nociceptive pressure compared to people in the low psychopathy group. However, SCR to pressure was similar in both groups. Our results suggest high psychopathic traits relate to problems with empathising with others' pain, as well as to reporting lower pressure intensities. Findings are also discussed in the context of the dual harm model.

Based on the idea of somatic aphasia, we expected to find significant differences in responses to nociceptive pressure in both self-report measures and SCRs. After undergoing the same matched procedure to select individualised moderate pressure stimulation levels, higher levels of nociceptive pressure were rated as more intense than touch/no pressure overall, as well as between psychopathy groups; those higher in psychopathy self-reported experiencing less nociceptive pressure than those lower in psychopathy. Conversely, there was a significant difference in SCR to high levels of nociceptive pressure, but not between psychopathy groups. Contrary to our findings, previous research has shown those with high levels of psychopathy may feel their own physical nociceptive pain experiences in a similar way as those lower in psychopathy, but their evaluations of the experience could be disconnected with their objective

sensations; this is known as somatic aphasia [35]. As a result, nociceptive pain is self-reported as less intense. Yet, people higher in psychopathy did not choose significantly higher levels of pressure when selecting their individualised pressure thresholds; the high psychopathy group self-reported lower levels of nociceptive pressure overall which indicates a difference in NRS self-reported pressure, but not necessarily a difference in nociceptive pressure perception itself. Furthermore, inconsistencies between physiological and self-reported data have been found. Participants higher in psychopathy scores reported similar scores to those lower in psychopathy when viewing negative images but showed reduced physiological activity to those images [34, 48]. A potential brain-body disconnect could be at play [e.g. 35], or deception may have been used since it is a central feature of a psychopathic personality [3]. Taken together, our findings did not support previous research suggesting that a lack of awareness or sensitivity to one's' own body sensations could underlie impairments in emotion in people with high psychopathic traits [35, 49]. Future research should examine this link more closely.

Previous studies have found that higher levels of psychopathy relate to reduced physiological responses to pain or fear [31, 34, 50], however, we did not find this for SCRs to mechanical pneumatic pressure. Although research is somewhat limited [51], our finding is surprising. Fear of a stimuli usually develops from past negative experiences [52], including pain experiences, which teach people to avoid pain-inducing stimuli. Yet, people with psychopathy usually do not associate pain with fear or punishment [53], hence they experience low physiological arousal [54, 55]. Alternatively, people with psychopathy may not interpret their body signals correctly. Research has found a relationship between psychopathy and difficulty in describing feelings [see 4 for meta-analysis] such as shame [56]. This is known as alexithymia [57]. Being unable to correctly identify feelings could lead to misinterpretation, and a different emotion is perceived [58]. Although our finding was non-significant, the link between alexithymia and psychopathic traits should be explored further, as previous research has found strong support for this relationship [e.g. 4].

We found that people in the high psychopathy group had different pain empathy reactions compared to people in the low psychopathy group. Firstly, our high psychopathy group self-reported feeling less empathy to images of other people's pain compared to the low psychopathy group. This is consistent with previous research which found people with high levels of psychopathy struggled to empathise with the moods and feelings of others [59], as well as a prison population's poor recognition of fear and disgust in others [60], and poor recognition of pain in young males with callous-unemotional traits [61]. Being able to recognise and respond to the emotions of others is important for social interactions [62, 63], thus, an absence of empathy for other people's pain could offer an explanation to why psychopathy relates to acts of violence that are carried out on others [59, 64]. Targeting empathy deficits in psychopathy, and in particular deficits for other people's pain could be an important strategy when creating interventions. For example, treatment programs should include teaching people to recognise painful situations and empathising with the pain of others.

Secondly, our high psychopathy group experienced less SCR to other people's pain images compared to our low psychopathy group, demonstrating less emotional arousal when observing others in pain. Our results are partially in line with previous findings which showed psychopathic inmates to have a reduction in SCR to the pain of others [34]. Our results demonstrate a lack of arousal to empathy exists outside of incarcerated samples and can apply to the general population, too. This is important as it shows criminal samples in previous findings and non-criminal psychopaths in the current sample may share similar psychophysiological characteristics [65]. On the other hand, participants may have also experienced faster habituation to pain images which could have led to reduced SCRs [see 66, 67]. As such, pain habituation should be a key area for exploration within psychopathy research. Additionally,

further research should be conducted comparing criminal and non-criminal psychopaths to find similarities and differences between the two samples; this would help to disentangle why some people with psychopathy are imprisoned while others are not.

According to the dual harm model, psychopaths have a strong disposition for aggression towards themselves and others [15]. Due to this, we predicted that individuals with psychopathy would have reduced empathy for others' pain as well as require objectively more intense pressure stimuli to report the same subjective pain intensity, for which we found support. Emotional dysregulation (a potential explanation for dual harm) is said to be higher in individuals with psychopathy [68]. In addition, literature shows under- and over-regulation may lead to a failure to contain emotions, or avoiding or suppressing emotional experiences, respectively [68, 69]. As such, it generates a disregard for others' pain. Although our results help to show support for this theory, more research is needed as this exciting concept is in its infancy.

## Strengths and limitations

The limitations of the study must be considered. The stimuli used depicted hands and feet in painful or matching non-painful situations and may not generalise to other painful or distressing situations. Therefore, a variety of painful and distressing situations should be explored in future studies. In addition, empathy has a contextual component; people empathise better with same sex and same race individuals [70]. Thus, future research needs to make stimuli more diverse and representative. Further, it is difficult to know whether an increase or decrease in SCR is because of arousal or not. For example, an increase in SCR to images may indicate interest [71] as opposed to feeling empathy. As SCR can be an ambiguous physiological measure and is difficult to interpret without also having self-report responses, future research should consider adopting them to reduce obscurity. Moreover, due to the nature of psychopathic traits, those high in psychopathy may experience pain but may mask it to appear tough, rather than experiencing a dissociation with bodily sensations. As such, future research should focus on unpicking this problem. In addition, since prior research has used a range of pain delivery methods [7, 22, 23], future research should consider using different modes of pain stimuli as differences in pain reporting could be modality specific. The internal consistency of some measures was not satisfactory; therefore, impending research should focus on improving this. Lastly, although appropriate statistical tests were used to balance the small sample size, we suggest conducting an a priori power analysis and testing psychopathy groups within larger samples to increase statistical power.

However, the current study also has its strengths. Firstly, SCR is an ecologically valid physiological measure with coherence found in both laboratory and real-life settings [72]. This means emotions captured in laboratory settings are like those in every-day situations. Additionally, we used pneumatic mechanical pressure which, arguably, better emulates nociceptive pain experienced every day, e.g. a finger trapped in a door, than other experimental modalities (e.g. laser, electrical, cold pressor). This is an important ecologically valid method as opposed to pain research using modalities which are not commonly experienced day-to-day. We also used a community sample which helped show psychopathy exists within the general population and adds to literature heavily based on clinical/incarcerated samples. This is of high clinical importance and practical relevance since interventions cannot be created on prison samples alone as people in the general population also exhibit psychopathological traits.

**Conclusions/future directions.** In conclusion, our study provides support for the dual harm model by demonstrating diminished perception of pain in people higher in psychopathy. Additionally, we contribute to current literature by showing that high psychopathic traits relate to problems in empathising with others' pain. As a result, future psychopathy interventions should thus focus on recognising and empathising with pain.

## Supporting information

**S1 Fig. Diagram representing the layout of the experimental setup in the laboratory.**
(DOCX)

**S2 Fig. Timeline for event-related (ER) analysis for pain task.** *a*–onset of pressure probe event which lasts up to 4 seconds; *b*– 1–4 second time window when any increase (over 0.1 microsiemens) in SC was taken as onset (*d*) of an ER SCR; *c*–amplitude of ER SCR.
(DOCX)

**S3 Fig. Timeline for event-related (ER) analysis for empathy task.** *a*–presentation of image; *b*– 1–4 second time window when any increase (over 0.1 microsiemens) in SC was (*d*) of an ER SCR; *c*–amplitude of ER SCR.
(DOCX)

**S1 Dataset. Data accessible for the study.**
(XLSX)

## Author Contributions

**Conceptualization:** Sophie Alshukri, Victoria Blinkhorn, Luna Muñoz, Nicholas Fallon.

**Data curation:** Sophie Alshukri, Minna Lyons, Luna Muñoz, Nicholas Fallon.

**Formal analysis:** Sophie Alshukri, Luna Muñoz.

**Investigation:** Sophie Alshukri.

**Methodology:** Sophie Alshukri, Luna Muñoz, Nicholas Fallon.

**Project administration:** Sophie Alshukri.

**Resources:** Sophie Alshukri, Luna Muñoz, Nicholas Fallon.

**Software:** Sophie Alshukri, Luna Muñoz, Nicholas Fallon.

**Supervision:** Minna Lyons, Luna Muñoz, Nicholas Fallon.

**Validation:** Sophie Alshukri, Luna Muñoz, Nicholas Fallon.

**Visualization:** Sophie Alshukri, Luna Muñoz, Nicholas Fallon.

**Writing – original draft:** Sophie Alshukri.

**Writing – review & editing:** Sophie Alshukri, Minna Lyons, Victoria Blinkhorn, Luna Muñoz, Nicholas Fallon.

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
