## [Decision Letter · Decision Letter 0]

21 Nov 2023

PONE-D-23-10103Psychopathy, pain, and pain empathy: A psychophysiological study.PLOS ONE

Dear Dr. Alshukri,

Thank you for submitting your manuscript to PLOS ONE. After careful consideration, we feel that it has merit but does not fully meet PLOS ONE’s publication criteria as it currently stands. Therefore, we invite you to submit a revised version of the manuscript that addresses the points raised during the review process.

We look forward to receiving your revised manuscript.

Kind regards,

Vincenzo De Luca

Academic Editor

PLOS ONE

Journal Requirements:

Reviewers' comments:

Reviewer's Responses to Questions

**Comments to the Author**

1. Is the manuscript technically sound, and do the data support the conclusions?

Reviewer #1: Partly

Reviewer #2: Partly

Reviewer #3: Yes

Reviewer #4: Yes

Reviewer #5: No

2. Has the statistical analysis been performed appropriately and rigorously? 

Reviewer #1: No

Reviewer #2: No

Reviewer #3: Yes

Reviewer #4: Yes

Reviewer #5: No

3. Have the authors made all data underlying the findings in their manuscript fully available?

Reviewer #1: No

Reviewer #2: No

Reviewer #3: No

Reviewer #4: No

Reviewer #5: Yes

4. Is the manuscript presented in an intelligible fashion and written in standard English?

Reviewer #1: Yes

Reviewer #2: No

Reviewer #3: Yes

Reviewer #4: No

Reviewer #5: Yes

5. Review Comments to the Author

Reviewer #1: The topic of the manuscript concerns the relationship between own pain, pain in others and psychopathic traits. There are strengths to the manuscript, but also major issues that need to be addressed. I have outlined them below.

INTRODUCTION

1) The writing is good in general, but the organisation of the introduction can be improved. I do not intend to sound harsh, but I was sometimes left with the impression that this was initially a thesis that was turned into an article. Adding more depth to the introduction and also logical transitions between paragraphs would help improve the introduction.

2) First sentence: Psychopathy is not a single personality trait, but a constellation of multiple traits. In general, the operationalization of the construct needs to be more elaborate, especially given that many of the studies cited in the manuscript use other operationalizations and measurements of psychopathy. Also, the idea that such traits can be measured in both criminal and non-criminal population needs to be explained clearly.

3) The manuscript would benefit greatly from a much more thorough coverage of research on pain empathy and psychopathy. Several key references are lacking and not discussed. For example, the study by Brazil, Atanassova & Oosterman (2022) addressed this same topic and is not even mentioned, even though it is perhaps the only published study to date examining links between psychopathy, own pain experience and experience of pain in others. The work by Bird and Viding (2014) is also highly relevant, as it provides a clear theory on how own emotions are the basis of empathy, which was explicitly linked to psychopathy (now the authors extrapolate claims from alexithymia research to say this psychopathy). This would help make the conceptual embedding offered in the introduction (and the discussion) much richer, thorough and balanced.

4) p3: '..it would be interesting to see how an individual’s psychopathy level relates to directly experienced pain...'. Whether something is interesting or not is subjective and not a solid ground for research. This makes the rationale for conducting the study less compelling. The authors should provide an argument based on empirical grounds instead. This requires a firmer embedding in prior research to identify the gaps of knowledge.

5) There are multiple claims made that are innacurate. For example, '...it has rarely been used to investigate the links between pain and psychopathy.' Such statements are not entirely accurate. There are definitely studies using psychophysiology (and even MRI) to assess own pain and/or pain in social context in relation to psychopathy. The authors should conduct a much more thorough search for articles and find a way to incorporate the papers they find into the manuscript to add nuance and depth and use them to articulate their reasoning more clearly. There is no need to be overly selective in discussing and citing studies. Some examples for where to start looking are:

Hare, R. D. (1965). Psychopathy, fear arousal and anticipated pain. Psychological Reports, 16, 499–502.

Marcoux, L.-A., Michon, P.-E., Voisin, J. I., Lemelin, S., Vachon-Presseau, E., & Jackson, P. L. (2013). The modulation of somatosensory resonance by psychopathic traits and empathy. Frontiers in Human Neuroscience, 7. https://doi.org/10.3389/fnhum.2013.00274

Brazil, I. A., Atanassova, D. V., & Oosterman, J. M. (2022). Own Pain Distress Mediates the Link Between the Lifestyle Facet of Psychopathy and Estimates of Pain Distress in Others. Frontiers in Behavioral Neuroscience, 16. https://www.frontiersin.org/article/10.3389/fnbeh.2022.824697

Decety, J., Skelly, L. R., & Kiehl, K. A. (2013). Brain Response to Empathy-Eliciting Scenarios Involving Pain in Incarcerated Individuals With Psychopathy. JAMA Psychiatry, 70(6), 638–645. https://doi.org/10.1001/jamapsychiatry.2013.27

Durand, G., & Plata, E. M. (2017). The effects of psychopathic traits on fear of pain, anxiety, and stress. Personality and Individual Differences, 119, 198–203. https://doi.org/10.1016/j.paid.2017.07.024

Heck, V., H, C., Driessen, J. M. A., Amato, M., Berg, V. D., N, M., Bhandari, P., Bilbao-Broch, L., Farres-Casals, J., Hendriks, M., Jodzio, A. C., Luque-Ballesteros, L., Schöchl, C., Velasco-Angeles, L. R., Weijer, R. H. A., Rijn, V., M, C., & Jongsma, M. L. A. (2017). Pain Processing in a Social Context and the Link with Psychopathic Personality Traits—An Event-Related Potential Study. Frontiers in Behavioral Neuroscience, 11. https://doi.org/10.3389/fnbeh.2017.00180

Seara-Cardoso, A., Viding, E., Lickley, R. A., & Sebastian, C. L. (2015). Neural responses to others’ pain vary with psychopathic traits in healthy adult males. Cognitive, Affective, & Behavioral Neuroscience, 15(3), 578–588. https://doi.org/10.3758/s13415-015-0346-7

Fedora, O., & Reddon, J. R. (1993). Psychopathic and nonpsychopathic inmates differ from normal controls in tolerance levels of electrical stimulation. Journal of Clinical Psychology, 49(3), 326–331.

METHODS

6) One of the bigger issues is that a series of t-tests were conducted to look for statistical group differences (and without any corrections). Could the authors explain why this approach was chosen, given the chance of obtaining false statistical results and the many problems with frequentist statistics? Wouldn't the use of approaches like MANOVA or Mixed Models be a better choice? It seems strange to discuss small sample size in the limitations section but not actively try to use statistical approaches that may be better in this case. If the findings are robust and reliable, they should remain so across statistical approaches. What would be even more convincing is that the findings remain the same even when the statistical approaches come from different statistical families (e.g., Robust statistics, Bayesian statistics). I request the authors to provide such information to ascertain the reader that the findings are reliable.

7) Sample size calculations are lacking.

8) The explanation of the empathy task on page 7 is very unclear. The authors should merge it with the task description that comes later to put all the information together.

DISCUSSION

9) p16: "..we also show that criminal and non-criminal psychopaths share similar physiological characteristics." This claim is misleading, as there were no criminal psychopaths included in the study. Moreover, the idea that psychopathy is a dimensional construct that can be measured in both criminal and non-criminal population is not explained in the manuscript, which is another indication that the theoretical embedding needs to be much more thorough.

10) Because of the poor embedding in prior research on pain in psychopathy, the discussion seems incomplete and superficial. I also noted that studies in youths are often used to develop arguments in the manuscript, even though there are plenty of studies in adults, who are more similar to the sample studied here. This creates a missed opportunity to position this study more firmly in the literature and highlight its added value.

GENERAL

11) I could not find information about data sharing and access, even though the journal requires this to be stated clearly.

Reviewer #2: Thank you for your submission of your article “Psychopathy, pain, and pain empathy: A psychophysiological study" to PLOSE ONE. In the present paper you intended to investigate experienced self-reported as well as psychophysiological pain and empathy for other's pain in participants showing low vs. high psychopathy. I believe that the investigated topic is of value but I have some major concerns about the manuscript. In my opinion, the article does not meet the standards of publication in PLOS ONE in its current state, therefore I recommend a thorough revision before the article is resubmitted to another journal.

First, the article lacks of a clear, systematic and statistical reasonable argumentation in the introduction as well as in the discussion. The line of argumentation is indeed supported with examples from (partly also not so current) literature. However, the described earlier correlations are presented without any statistical indicators. The associations or correlations could be much better interpreted by reporting effect sizes. At times, the argumentation is also unclear and ambiguously explained (e.g., painful facial expression as the ultimate way to show they need help). Furthermore, it is not clear how the results of the present study could explain characteristics of non-criminal and criminal individuals, as this socio-demographic feature was not a part of the study design at all.

Second, and what is particularly concerning, the article is not based on any sound statistical methods. Foremost, the achieved scores of the n = 131 participants, forming the basis of the final sample, were not reported. Therefore, it is not comprehensible how to differentiate the top 20% and lowest values. It could be possible that, with a 4-point Likert scale, as used in this questionnaire, the values were very closely distributed, and the excluded individuals may have hardly differed from those considered as high or low psychopathic participants. Additionally, socio-demographic characteristics would also be of interest here.

The applied method relied solely on t-tests and the assessment of statistical significance. With respect to the new statistics, it is highly recommended to not only evaluate results based on significance (dichotomous, yes or no) but to interpret effect sizes. Moreover, even the chosen alpha level was not specified. Although Cohen ds were reported in later sections (where, given the very small sample size (another point), Hedges' g would be more advisable as this effect size corrects for small samples), the calculation of these standardized mean differences was not described, nor were any thresholds mentioned for interpreting these results (i.e., small/large/medium effects, according to Cohen, 1988). Furthermore, although these Cohen ds were reported in the text and tables, they were not interpreted. In my opinion, this approach is inadequate.

Finally, the present study was neither preregistered nor is the analysis code or data publicly accessible. Although the authors state that the data would be made available upon request, this approach is also highly concerning with regards to Open Science initiatives.

I hope that these comments were helpful for the future work on the study and I wish the authors good luck with a resubmission!

Reviewer #3: The present manuscript investigates the relationship between psychopathy, pain and empathy using both self-reports and psychophysiological measures. The results indicate less self-reported physical pain and empathy in people higher in psychopathy, compared to those lower in psychopathy. Notably, the results obtained using skin conductance response (SCR) to physical pain did not point to differences between people residing in highest and lowest fifth of the psychopathy spectrum.

These main results of the study are highly significant for the discipline as they suggest distinct evaluation of pain in people with psychopathic traits, with only subjective experiences of pain distinguishing people higher and lower in psychopathy. Regarding empathy, both objective and subjective measures suggested distinct responses in people residing in highest and lowest fifth of the psychopathy spectrum.

Both the identified research gaps (see Introduction) and the findings of the study (see Discussion) are properly placed in the context of the previous literature, considering a wide range of studies conducted in the field. Furthermore, the data and analyses fully support the authors’ conclusions. The procedure along with the utilised measures and analyses is described in sufficient detail, ensuring transparency and comprehensibility of the methods used. Finally, the authors discuss the implications of these findings in an intelligible fashion and propose several ideas for future research.

Taken together, the manuscript is well organized, clearly written and scientifically sound. However, there are some points that should be addressed prior to accepting the paper for publication. For details, please see the attachment.

Reviewer #4: This experimental study assessed self-reported pain and SCR during (a) self-experienced pressure pain stimulation and (b) while watching pictures of others in pain, comparing results between two groups of students distinguished by their high versus low levels of psychopathic trait. The results show lower reported empathic pain and SCR in the empathy condition, while in the self pain condition the high psychopathic group reported lower high pain but no difference for low pain or for SCR in either pain intensity condition. The topic is interesting. The methods appear to be well done as far as I can see, although some information is missing here (see below).

I have the following main concerns about this study:

1) A main problem is the small sample size – only 25-26 participants in each group.

2) The analysis plan and results presentation is overall rather confusing. This may be due to the authors failing to make 2 important distinctions in their terminology: First, SCR does not measure pain, but physiological activity that may be related to pain. Pain is by definition a subjective experience. Second, the reported pain intensities need to be distinguished from the physical stimulus intensities (pressure) that were used to elicit that pain intensity. As it is, I am unsure throughout the paper what the authors are referring to.

Another problem with the results (but maybe related to the above point) is that all self-experienced pain stimuli were individually adjusted for each participant to low (3/10) or high (6-7/10) intensity. Given this individual adjustment, I do not understand why the authors expected differences in pain ratings between the groups, and how they interpret them. My only idea would be faster habituation in the high psychopathic group, but this topic is not discussed anywhere. Would not it make more sense to compare the physical pressure intensities in this setup? Also it would be helpful to present the results visually (one could skip the SCR timing images if necessary).

3) There is already quite some literature in relation to psychopathy on the topics of SCR, pain and empathy. The presentation of this literature appears to be insufficient. For example, I am missing the article by Meffert et al. (2015) showing that pychopathic individuals can show relatively normal empathy when explicitly asked to feel with the other person. The authors need to explain more convincingly what is new about their study, and also explain more clearly what their hypotheses are and why.

4) Some information is missing on methods: E.g. what electrodes and what gel was used to measure SCR? How was the SCR signal standardized to be comparable between participants (Boucsein et al. 2012)?

5) The manuscript seems to contain many typos and there are some problems with the English language throughout.

Minor points:

Abstract:

The abstract should contain more methods details, e.g. names of the questionnaires, used pain methods.

One of the last sentences (lack of awareness to ones’ body sensation may underlie emotional impairments in psychopathy) seems like a new hypothesis, not a conclusion that can be drawn from the results of this study.

Introduction:

p. 3: The sentence “This raises the question of whether there is a link between psychopathy, empathy for others, and experiencing pain“ seems to give a wrong impression – there is already quite some research on the topic of psychopthy, pain perception and empathy.

p. 4: Why do self-report measures reduce the likeliness of socially desirable responding?

The hypotheses mentioned at the end of the introduction do not follow logically from the introduction and are confusing because it is not clarified whether they refer to self-report or SCR results. Given the introduction one would rather expect a hypothesis on the relation between self-report and SCR results.

Methods:

p. 5: Sample: The authors screened N=369 students for psychopathy. How is it possible that as many as N=131 students fulfilled the criterion of upper/lower 20% of psychopathy spectrum?

p. 6: Pain test: a VAS does not have numbers/ticks in between, but a SAM does. Please clarify if you used a SAM / NRS (numerical rating scale) or a VAS.

p. 10: Empathy task: Why was a different pain scale used compared to the pain task (different range 0-100, different end point descriptors)?

Please clarify and credit who created the pictures. Where they created by Fallon et al. (2015)?

Results:

p. 13: Why are average pain results >10 when pain intensity was assessed on a 0-10 scale (see methods)?

p. 14: Manipulation checks: A non-significant trend is presented as if it were a sign result.

Discussion:

The conclusion that the high psychopathy group reported lower self-experienced pain is true only for the high pain condition, and the meaning of this result remains unclear. It should be clarified (already in the methods) whether the low pain condition is actually a no pain condition.

p. 16: “Yet, people higher in psychopathy did not choose significantly higher levels of pain when selecting their pain thresholds…” the phrase suggests that the authors failed to distinguish pain intensity (a subjective rating) from physical stimulus pressure intensity (an objective rating).

References:

Boucsein W, Fowles DC, Grimnes S, Ben-Shakhar G, roth WT, Dawson ME, Filion DL; Society for Psychophysiological Research Ad Hoc Committee on Electrodermal Measures. Publication recommendations for electrodermal measurements. Psychophysiology. 2012 Aug;49(8):1017-34. doi: 10.1111/j.1469-8986.2012.01384.x. Epub 2012 Jun 8. PMID: 22680988.

Harma Meffert, Valeria Gazzola, Johan A. den Boer, Arnold A. J. Bartels, Christian Keysers, Reduced spontaneous but relatively normal deliberate vicarious representations in psychopathy, Brain, Volume 136, Issue 8, August 2013, Pages 2550–2562, https://doi.org/10.1093/brain/awt190

Reviewer #5: This paper examines whether people scoring high and low on the psychopathy scale differ in their SCR to pain stimulus, their self-reports on pain, and whether high and low people score differently when it comes to empathizing with other’s pain.

This research has major flaws regarding study design, empirical strategy, and statistical analysis. I only list the major issues here:

1. Just because someone who scores high on the psychopathy scale reports low self-reported pain, we cannot conclude that these people have dissociation between their bodily sensations and what they experience. This self-report is confounded with their high psychopathy level, which can also impact their image concerns and their manipulative motives. E.g., even though they experience high pain, they will not report it to be seen as tough.

2. The sample size is very small for all these pairwise analyses. There is no sample size calculation presented.

3. There are at least ten pairwise comparisons in the paper and no adjustment for them. Bonferroni correction would need to adjust alfa – level to 0.05/10 which would eliminate most of the results.

4. Weird that the reported degrees of freedom for the independent sample t-tests are sometimes 47 (which is okay, given the 26 and 23 participants in the two groups) and sometimes 45 (which is strange).

5. I could not reproduce the t-test statistics based on the reported means and SDs.

6. Strange that in some cases SE and not SDs for means are reported and yet independent t-statistics are presented.

7. Based on the reported means, SDs and Ns, I could not reproduce the reported p-values.

6. PLOS authors have the option to publish the peer review history of their article (what does this mean?). If published, this will include your full peer review and any attached files.

Reviewer #1: No

Reviewer #2: No

Reviewer #3: No

Reviewer #4: No

Reviewer #5: No

---

## [Author Response · Author response to Decision Letter 0]

9 Apr 2024

Dear reviewers,

Thank you very much for the comments you have made on our manuscript. We appreciate the feedback and have hopefully addressed all the comments that have been made to the correct standard. Below, you will find each of the comments that we received and the action we have taken. 

Thank you for your time and effort when reviewing our work.

5. Review Comments to the Author

Reviewer #1: The topic of the manuscript concerns the relationship between own pain, pain in others and psychopathic traits. There are strengths to the manuscript, but also major issues that need to be addressed. I have outlined them below.

INTRODUCTION

1) The writing is good in general, but the organisation of the introduction can be improved. I do not intend to sound harsh, but I was sometimes left with the impression that this was initially a thesis that was turned into an article. Adding more depth to the introduction and also logical transitions between paragraphs would help improve the introduction.

 Response: Thank you for your comment. We have added more depth and logical transitions to the introduction.

2) First sentence: Psychopathy is not a single personality trait, but a constellation of multiple traits. In general, the operationalization of the construct needs to be more elaborate, especially given that many of the studies cited in the manuscript use other operationalizations and measurements of psychopathy. Also, the idea that such traits can be measured in both criminal and non-criminal population needs to be explained clearly.

 Response: Thank you for your comment. We have expanded our reading and added more research to help support our argument. 

3) The manuscript would benefit greatly from a much more thorough coverage of research on pain empathy and psychopathy. Several key references are lacking and not discussed. For example, the study by Brazil, Atanassova & Oosterman (2022) addressed this same topic and is not even mentioned, even though it is perhaps the only published study to date examining links between psychopathy, own pain experience and experience of pain in others. The work by Bird and Viding (2014) is also highly relevant, as it provides a clear theory on how own emotions are the basis of empathy, which was explicitly linked to psychopathy (now the authors extrapolate claims from alexithymia research to say this psychopathy). This would help make the conceptual embedding offered in the introduction (and the discussion) much richer, thorough and balanced.

Response: Thank you for your comment. Brazil et al (2022) and Bird & Viding (2014) have been read and included in the introduction to help the conceptual embedding. 

4) p3: '..it would be interesting to see how an individual’s psychopathy level relates to directly experienced pain...'. Whether something is interesting or not is subjective and not a solid ground for research. This makes the rationale for conducting the study less compelling. The authors should provide an argument based on empirical grounds instead. This requires a firmer embedding in prior research to identify the gaps of knowledge.

Response: Wording of the last sentence changed. 

5) There are multiple claims made that are innacurate. For example, '...it has rarely been used to investigate the links between pain and psychopathy.' Such statements are not entirely accurate. There are definitely studies using psychophysiology (and even MRI) to assess own pain and/or pain in social context in relation to psychopathy. The authors should conduct a much more thorough search for articles and find a way to incorporate the papers they find into the manuscript to add nuance and depth and use them to articulate their reasoning more clearly. There is no need to be overly selective in discussing and citing studies. Some examples for where to start looking are:

Response: Statement containing “rarely been used” amended. Papers recommended have been read to increase breadth and depth of knowledge, and some have been included in the manuscript.

Hare, R. D. (1965). Psychopathy, fear arousal and anticipated pain. Psychological Reports, 16, 499–502.

Marcoux, L.-A., Michon, P.-E., Voisin, J. I., Lemelin, S., Vachon-Presseau, E., & Jackson, P. L. (2013). The modulation of somatosensory resonance by psychopathic traits and empathy. Frontiers in Human Neuroscience, 7. https://doi.org/10.3389/fnhum.2013.00274

Brazil, I. A., Atanassova, D. V., & Oosterman, J. M. (2022). Own Pain Distress Mediates the Link Between the Lifestyle Facet of Psychopathy and Estimates of Pain Distress in Others. Frontiers in Behavioral Neuroscience, 16. https://www.frontiersin.org/article/10.3389/fnbeh.2022.824697

Decety, J., Skelly, L. R., & Kiehl, K. A. (2013). Brain Response to Empathy-Eliciting Scenarios Involving Pain in Incarcerated Individuals With Psychopathy. JAMA Psychiatry, 70(6), 638–645. https://doi.org/10.1001/jamapsychiatry.2013.27

Durand, G., & Plata, E. M. (2017). The effects of psychopathic traits on fear of pain, anxiety, and stress. Personality and Individual Differences, 119, 198–203. https://doi.org/10.1016/j.paid.2017.07.024

Heck, V., H, C., Driessen, J. M. A., Amato, M., Berg, V. D., N, M., Bhandari, P., Bilbao-Broch, L., Farres-Casals, J., Hendriks, M., Jodzio, A. C., Luque-Ballesteros, L., Schöchl, C., Velasco-Angeles, L. R., Weijer, R. H. A., Rijn, V., M, C., & Jongsma, M. L. A. (2017). Pain Processing in a Social Context and the Link with Psychopathic Personality Traits—An Event-Related Potential Study. Frontiers in Behavioral Neuroscience, 11. https://doi.org/10.3389/fnbeh.2017.00180

Seara-Cardoso, A., Viding, E., Lickley, R. A., & Sebastian, C. L. (2015). Neural responses to others’ pain vary with psychopathic traits in healthy adult males. Cognitive, Affective, & Behavioral Neuroscience, 15(3), 578–588. https://doi.org/10.3758/s13415-015-0346-7

Fedora, O., & Reddon, J. R. (1993). Psychopathic and nonpsychopathic inmates differ from normal controls in tolerance levels of electrical stimulation. Journal of Clinical Psychology, 49(3), 326–331.

METHODS

6) One of the bigger issues is that a series of t-tests were conducted to look for statistical group differences (and without any corrections). Could the authors explain why this approach was chosen, given the chance of obtaining false statistical results and the many problems with frequentist statistics? Wouldn't the use of approaches like MANOVA or Mixed Models be a better choice? It seems strange to discuss small sample size in the limitations section but not actively try to use statistical approaches that may be better in this case. If the findings are robust and reliable, they should remain so across statistical approaches. What would be even more convincing is that the findings remain the same even when the statistical approaches come from different statistical families (e.g., Robust statistics, Bayesian statistics). I request the authors to provide such information to ascertain the reader that the findings are reliable.

Response: Thank you for your suggestion for using a MANOVA test. On consideration of this, we decided that this would not be suitable as we only have one independent variable: psychopathy groups (low and high). Instead, we conducted independent sample’s t-tests using Hedge’s g effect sizes and confidence intervals to correct for our small sample sizes.

7) Sample size calculations are lacking.

 Response: Thank you for your comment. We unfortunately did not conduct a sample size calculation prior to the research however we have commented on this in the limitations section of the manuscript and recommended future research do this. 

8) The explanation of the empathy task on page 7 is very unclear. The authors should merge it with the task description that comes later to put all the information together.

Response: Thank you for your comment. Both sections have been combined into empathy task description.

DISCUSSION

9) p16: "..we also show that criminal and non-criminal psychopaths share similar physiological characteristics." This claim is misleading, as there were no criminal psychopaths included in the study. Moreover, the idea that psychopathy is a dimensional construct that can be measured in both criminal and non-criminal population is not explained in the manuscript, which is another indication that the theoretical embedding needs to be much more thorough.

Response. Thank you for your comment. The claim comparing criminal and non-criminal psychopaths has been removed, and instead has been used as a discussion point only.

10) Because of the poor embedding in prior research on pain in psychopathy, the discussion seems incomplete and superficial. I also noted that studies in youths are often used to develop arguments in the manuscript, even though there are plenty of studies in adults, who are more similar to the sample studied here. This creates a missed opportunity to position this study more firmly in the literature and highlight its added value.

Response: Thank you for your comment. A more appropriate adult sample study has been used instead.

GENERAL

11) I could not find information about data sharing and access, even though the journal requires this to be stated clearly.

Response: Thank you for your comment. Data will be shared once the reviewer comments have been completed and the manuscript has been resubmitted. 

Reviewer #2: Thank you for your submission of your article “Psychopathy, pain, and pain empathy: A psychophysiological study" to PLOSE ONE. In the present paper you intended to investigate experienced self-reported as well as psychophysiological pain and empathy for other's pain in participants showing low vs. high psychopathy. I believe that the investigated topic is of value but I have some major concerns about the manuscript. In my opinion, the article does not meet the standards of publication in PLOS ONE in its current state, therefore I recommend a thorough revision before the article is resubmitted to another journal.

First, the article lacks of a clear, systematic and statistical reasonable argumentation in the introduction as well as in the discussion. The line of argumentation is indeed supported with examples from (partly also not so current) literature. However, the described earlier correlations are presented without any statistical indicators. The associations or correlations could be much better interpreted by reporting effect sizes. At times, the argumentation is also unclear and ambiguously explained (e.g., painful facial expression as the ultimate way to show they need help). Furthermore, it is not clear how the results of the present study could explain characteristics of non-criminal and criminal individuals, as this socio-demographic feature was not a part of the study design at all.

Second, and what is particularly concerning, the article is not based on any sound statistical methods. Foremost, the achieved scores of the n = 131 participants, forming the basis of the final sample, were not reported. Therefore, it is not comprehensible how to differentiate the top 20% and lowest values. It could be possible that, with a 4-point Likert scale, as used in this questionnaire, the values were very closely distributed, and the excluded individuals may have hardly differed from those considered as high or low psychopathic participants. Additionally, socio-demographic characteristics would also be of interest here.

The applied method relied solely on t-tests and the assessment of statistical significance. With respect to the new statistics, it is highly recommended to not only evaluate results based on significance (dichotomous, yes or no) but to interpret effect sizes. Moreover, even the chosen alpha level was not specified. Although Cohen ds were reported in later sections (where, given the very small sample size (another point), Hedges' g would be more advisable as this effect size corrects for small samples), the calculation of these standardized mean differences was not described, nor were any thresholds mentioned for interpreting these results (i.e., small/large/medium effects, according to Cohen, 1988). Furthermore, although these Cohen ds were reported in the text and tables, they were not interpreted. In my opinion, this approach is inadequate.

Response: Thank you for your comment. We have added the minimum and maximum values and means and standard deviations of the YPI that participants completed for the low and high psychopathy group to show the difference between the two groups’ scores. In addition, Hedge’s g for sample size has been used to correct for our small sample size. 

Finally, the present study was neither preregistered nor is the analysis code or data publicly accessible. Although the authors state that the data would be made available upon request, this approach is also highly concerning with regards to Open Science initiatives.

Response: Thank you for your comment. Data will be shared once the manuscript has been resubmitted.

I hope that these comments were helpful for the future work on the study and I wish the authors good luck with a resubmission!

Reviewer #3: The present manuscript investigates the relationship between psychopathy, pain and empathy using both self-reports and psychophysiological measures. The results indicate less self-reported physical pain and empathy in people higher in psychopathy, compared to those lower in psychopathy. Notably, the results obtained using skin conductance response (SCR) to physical pain did not point to differences between people residing in highest and lowest fifth of the psychopathy spectrum.

These main results of the study are highly significant for the discipline as they suggest distinct evaluation of pain in people with psychopathic traits, with only subjective experiences of pain distinguishing people higher and lower in psychopathy. Regarding empathy, both objective and subjective measures suggested distinct responses in people residing in highest and lowest fifth of the psychopathy spectrum.

Both the identified research gaps (see Introduction) and the findings of the study (see Discussion) are properly placed in the context of the previous literature, considering a wide range of studies conducted in the field. Furthermore, the data and analyses fully support the authors’ conclusions. The procedure along with the utilised measures and analyses is described in sufficient detail, ensuring transparency and comprehensibility of the methods used. Finally, the authors discuss the implications of these findings in an intelligible fashion and propose several ideas for future research.

Taken together, the manuscript is well organized, clearly written and scientifically sound. However, there are some points that should be addressed prior to accepting the paper for publication. For details, please see the 

attachment.

Taken together, the manuscript is well organized, clearly written and scientifically sound. However, there are some points that should be addressed prior to accepting the paper for publication:

Abstract:

● Consider replacing “psychophysiology” with “psychophysiological measures” in the first paragraph.

Response: Thank you for your comment. This change has been made.

● Consider replacing “We suggest psychopathy interventions focus on recognising and empathising with pain.” with “Psychopathy interventions should thus focus both on recognising and empathising with pain.”

Response: Thank you for your comment. This change has been made. 

Introduction:

● Please add a reference after the first sentence.

Response: Reference added.

● Page 3: Instead of “Why look into pain?”, use the following wording: “There are several

reasons for investigating pain in the context of psychopathy.” to ensure a smooth transition

between the first and second paragraph.

Response: Thank you for the comment. This change has been made.

● Page 4: Since Plos One is 

---

## [Decision Letter · Decision Letter 1]

26 Apr 2024

PONE-D-23-10103R1Psychopathy, pain, and pain empathy: A psychophysiological study.PLOS ONE

Dear Dr. Alshukri,

Thank you for submitting your manuscript to PLOS ONE. After careful consideration, we feel that it has merit but does not fully meet PLOS ONE’s publication criteria as it currently stands. Therefore, we invite you to submit a revised version of the manuscript that addresses the points raised during the review process.

We look forward to receiving your revised manuscript.

Kind regards,

Vincenzo De Luca

Academic Editor

PLOS ONE

Reviewers' comments:

Reviewer's Responses to Questions

**Comments to the Author**

1. If the authors have adequately addressed your comments raised in a previous round of review and you feel that this manuscript is now acceptable for publication, you may indicate that here to bypass the “Comments to the Author” section, enter your conflict of interest statement in the “Confidential to Editor” section, and submit your "Accept" recommendation.

Reviewer #2: All comments have been addressed

Reviewer #3: (No Response)

Reviewer #4: (No Response)

2. Is the manuscript technically sound, and do the data support the conclusions?

Reviewer #2: Yes

Reviewer #3: Yes

Reviewer #4: Partly

3. Has the statistical analysis been performed appropriately and rigorously? 

Reviewer #2: Yes

Reviewer #3: Yes

Reviewer #4: No

4. Have the authors made all data underlying the findings in their manuscript fully available?

Reviewer #2: Yes

Reviewer #3: Yes

Reviewer #4: No

5. Is the manuscript presented in an intelligible fashion and written in standard English?

Reviewer #2: Yes

Reviewer #3: Yes

Reviewer #4: No

6. Review Comments to the Author

Reviewer #2: The authors improved their manuscript and addressed the reviewer's comments appropriately. Please italicize statistical expressions in the methods section.

Reviewer #3: Abstract:

• When introducing the abbreviation YPI, write out the full term first and then write the abbreviations in the parenthesis.

Introduction:

• Line 53: Replace “stem” with “stems” as it refers to inability (i.e. inability stems).

• Line 62-63: The part “however, when asked to empathise…” is unclear. I think this is because a subject is missing. Please revise.

• Line 86: The sentence should be rephrased – measures cannot correlate. Did you mean: “Psychopathy was correlated with increased pain tolerance in studies using self-report measures.”? Please check.

• Line 120: I suggest adding a reference for “dual harm model”.

Methods:

• Line 237: The opening quantitation mark is missing in the example item for impulsiveness.

• Line 242: This indention is bigger that others.

• Line 268-269: Please use italics for subsample size (n).

• Line 278: I think “high” should be replaces with “higher”.

Results:

• Line 287: I suggest putting (Z = 1.94) after “directly experienced pain”.

• Line 301: “Ehen given higher levels of pain” should be deleted as this part already appears at the beginning of the sentence.

• Line 348: “images” should be deleted after “pain”.

Discussion:

• Line 362: Replace “its” with “their” since you are referring to traits.

• Line 386: To improve readability, add “that” after “suggest”.

• Line 388-389: I find “other possible psychological areas” too general. Please choose a more specific term (e.g. other psychological areas related to pain and empathy).

• Line 457: Replace “were” with “was” since you are referring to internal consistency.

Reviewer #4: The literature review in the introduction is much improved and develops the research question more convincingly now. However, several of my other previous major comments have not been adressed sufficiently by the authors, including:

1) Wrong interpretation of the pain rating results: The authors compare self-pain ratings (NRS) between the groups for pressure stimuli whose intensity was individually adjusted so that they induced the same pain rating in all participants in the first part of the experiment. The finding that these high pain stimuli later induced lower pain ratings in high versus low psychopathic individuals (p = .03) does not indicate overall lower pain perception in high psychopathic individuals, but a reduction in pain perception across the experiment. This is an interesting result, possibly due to faster habituation or, alternatively, different reporting as high psychpathic individuals may choose not to acknowledge their pain as the experiment proceeds.

However, to answer the first main research question of the study – whether high psychopathic individuals experience less intense pain compared to low psychopathic individuals – the objective volts leading to similar perception of high pain during pain calibration are at least equally if not more relevant, given that participants are less likely to manipulate their answers during a random staircase stimulation session. These results are reported in Line 481-483 and show a non-significant trend (p = 0.05) in the right direction (lower pain sensitivity in high psychopathic individuals during pain calibration), but this is not a manipulation check but a major result, and it needs to be correctly presented and discussed.

2) In consideration of the small sample size, the authors should have replaced multiple t-tests by a single ANOVA, and should additionally have applied alpha-corrections to the post-hoc contracts following that ANOVA to render their findings more reliable, but this was not done in spite of reviewer comments.

3) Related to the above points, I do not agree with the main conclusion of the study: Even if the weak significant group difference regarding self-pain ratings found in the study would survive the more conservative approach of ANOVA + alpha correction, overall the main message from the data seems to be that group differences are evident in empathy much more than in self-perceived pain. This also speaks against the claim that lack of awareness of body sensations is the cause of low empathy in psychopaths.

4) Wrong terminology is still used in many places, e.g.

* No correct distinction between nociceptive stimuli (defined by objective volts) and pain (a subjective response to those stimuli) is creating confusion throughout the manuscript, e.g.

Line 586: „Yet, people higher in psychopathy did not choose significantly higher levels of pain when selecting their pain thresholds.“ should be rephrased to „...did not choose significantly higher levels of pressure when selecting their pain threshold“.

* SCR is still described as a measure of pain when in fact it is a measure of arousal, which can also be caused by stimuli of positive valence – a relevant point when showing pictures of others in pain to psychopaths. This problem is already evident in the way the research questions are phrased at the end of the introduction.

* The authors mention pain tolerance in several places, e.g. Line 639: „...we predicted that individuals with psychopathy would have… higher pain tolerance themselves“. Pain tolerance (i.e. the intensity of pressure and/or amount of time a painful stimulus is tolerated before aborting it) was not measured in this study. Only pain perception was measured.

7. PLOS authors have the option to publish the peer review history of their article (what does this mean?). If published, this will include your full peer review and any attached files.

Reviewer #2: No

Reviewer #3: No

Reviewer #4: No

---

## [Author Response · Author response to Decision Letter 1]

14 May 2024

Dear reviewers,

Thank you very much for the comments you have made on our manuscript. We appreciate the feedback and have hopefully addressed all the comments that have been made to the correct standard. Below, you will find each of the comments that we received and the action we have taken. 

Thank you for your time and effort when reviewing our work.

Reviewer #2: The authors improved their manuscript and addressed the reviewer's comments appropriately. Please italicize statistical expressions in the methods section.

 Response: Thank you for your comment and taking the time to review our manuscript. 

Reviewer #3: Abstract:

• When introducing the abbreviation YPI, write out the full term first and then write the abbreviations in the parenthesis.

 Response: Thank you for your comment. This change has been made. 

Introduction:

• Line 53: Replace “stem” with “stems” as it refers to inability (i.e. inability stems).

 Response: Thank you. Change has been made. 

• Line 62-63: The part “however, when asked to empathise…” is unclear. I think this is because a subject is missing. Please revise.

 Response: Thank you for your comment. We have revised this sentence. 

• Line 86: The sentence should be rephrased – measures cannot correlate. Did you mean: “Psychopathy was correlated with increased pain tolerance in studies using self-report measures.”? Please check.

 Response: Thank you for your comment. This change has been made. 

• Line 120: I suggest adding a reference for “dual harm model”.

Response: Thank you for your comment. This reference has been added. 

Methods:

• Line 237: The opening quantitation mark is missing in the example item for impulsiveness.

Response: This has been added. 

• Line 242: This indention is bigger that others.

Response: This has been amended. 

• Line 268-269: Please use italics for subsample size (n).

Response: Italics have been added. 

• Line 278: I think “high” should be replaces with “higher”.

Response: This change has been made. 

Results:

• Line 287: I suggest putting (Z = 1.94) after “directly experienced pain”.

Response: This change has been made. 

• Line 301: “Ehen given higher levels of pain” should be deleted as this part already appears at the beginning of the sentence.

Response: This has been removed.

• Line 348: “images” should be deleted after “pain”.

Response: This has been removed. 

Discussion:

• Line 362: Replace “its” with “their” since you are referring to traits.

Response: This has been changed.

• Line 386: To improve readability, add “that” after “suggest”.

Response: This has been added.

• Line 388-389: I find “other possible psychological areas” too general. Please choose a more specific term (e.g. other psychological areas related to pain and empathy).

Response: This has been added. 

• Line 457: Replace “were” with “was” since you are referring to internal consistency.

Response: This has been changed.

Reviewer #4: The literature review in the introduction is much improved and develops the research question more convincingly now. However, several of my other previous major comments have not been adressed sufficiently by the authors, including:

1) Wrong interpretation of the pain rating results: The authors compare self-pain ratings (NRS) between the groups for pressure stimuli whose intensity was individually adjusted so that they induced the same pain rating in all participants in the first part of the experiment. The finding that these high pain stimuli later induced lower pain ratings in high versus low psychopathic individuals (p = .03) does not indicate overall lower pain perception in high psychopathic individuals, but a reduction in pain perception across the experiment. This is an interesting result, possibly due to faster habituation or, alternatively, different reporting as high psychpathic individuals may choose not to acknowledge their pain as the experiment proceeds.

 Response: Thank you for your comment. Pain habituation has been addressed in the discussion, and the limitation of people with psychopathy not admitting to pain due to masking or wanting to appear tough. 

However, to answer the first main research question of the study – whether high psychopathic individuals experience less intense pain compared to low psychopathic individuals – the objective volts leading to similar perception of high pain during pain calibration are at least equally if not more relevant, given that participants are less likely to manipulate their answers during a random staircase stimulation session. These results are reported in Line 481-483 and show a non-significant trend (p = 0.05) in the right direction (lower pain sensitivity in high psychopathic individuals during pain calibration), but this is not a manipulation check but a major result, and it needs to be correctly presented and discussed.

Response: Thank you for your comment. Since running the ANOVA, this result is no longer in the right direction (towards significance) and so have not been regarded as a major result as previously suggested. 

2) In consideration of the small sample size, the authors should have replaced multiple t-tests by a single ANOVA, and should additionally have applied alpha-corrections to the post-hoc contracts following that ANOVA to render their findings more reliable, but this was not done in spite of reviewer comments.

Response: Thank you for your comment. We have ran the ANOVA to make findings more reliable, and altered the discussion based on those changes.

3) Related to the above points, I do not agree with the main conclusion of the study: Even if the weak significant group difference regarding self-pain ratings found in the study would survive the more conservative approach of ANOVA + alpha correction, overall the main message from the data seems to be that group differences are evident in empathy much more than in self-perceived pain. This also speaks against the claim that lack of awareness of body sensations is the cause of low empathy in psychopaths.

Response: Thank you for your comment. We conducted the recommended ANOVA and self-pain ratings did not survive this test, thus, this is no longer a conclusion of the study.

4) Wrong terminology is still used in many places, e.g.

* No correct distinction between nociceptive stimuli (defined by objective volts) and pain (a subjective response to those stimuli) is creating confusion throughout the manuscript, 

 Response: Thank you for your comment. Adjustments have been made when referring to nociceptive stimuli and subjective responses to pain.

e.g.

Line 586: „Yet, people higher in psychopathy did not choose significantly higher levels of pain when selecting their pain thresholds.“ should be rephrased to „...did not choose significantly higher levels of pressure when selecting their pain threshold“.

Response: Thank you for your comment. This has been addressed.

* SCR is still described as a measure of pain when in fact it is a measure of arousal, which can also be caused by stimuli of positive valence – a relevant point when showing pictures of others in pain to psychopaths. This problem is already evident in the way the research questions are phrased at the end of the introduction.

Response: Thank you for your comment. This change has been made.

* The authors mention pain tolerance in several places, e.g. Line 639: „...we predicted that individuals with psychopathy would have… higher pain tolerance themselves“. Pain tolerance (i.e. the intensity of pressure and/or amount of time a painful stimulus is tolerated before aborting it) was not measured in this study. Only pain perception was measured.

Response: Thank you for your comment. We have changed ‘pain tolerance’ to ‘pain perception/physical pain/greater levels of pain’.

---

## [Decision Letter · Decision Letter 2]

27 May 2024

PONE-D-23-10103R2Psychopathy, pain, and pain empathy: A psychophysiological study.PLOS ONE

Dear Dr. Alshukri,

Thank you for submitting your manuscript to PLOS ONE. After careful consideration, we feel that it has merit but does not fully meet PLOS ONE’s publication criteria as it currently stands. Therefore, we invite you to submit a revised version of the manuscript that addresses the points raised during the review process.

We look forward to receiving your revised manuscript.

Kind regards,

Vincenzo De Luca

Academic Editor

PLOS ONE

Reviewers' comments:

Reviewer's Responses to Questions

**Comments to the Author**

1. If the authors have adequately addressed your comments raised in a previous round of review and you feel that this manuscript is now acceptable for publication, you may indicate that here to bypass the “Comments to the Author” section, enter your conflict of interest statement in the “Confidential to Editor” section, and submit your "Accept" recommendation.

Reviewer #3: All comments have been addressed

Reviewer #4: (No Response)

2. Is the manuscript technically sound, and do the data support the conclusions?

Reviewer #3: Yes

Reviewer #4: No

3. Has the statistical analysis been performed appropriately and rigorously? 

Reviewer #3: Yes

Reviewer #4: I Don't Know

4. Have the authors made all data underlying the findings in their manuscript fully available?

Reviewer #3: Yes

Reviewer #4: No

5. Is the manuscript presented in an intelligible fashion and written in standard English?

Reviewer #3: Yes

Reviewer #4: Yes

6. Review Comments to the Author

Reviewer #3: It is clear that the authors made a considerable effort to incorporate reviewers‘ feedback. In my opinion, there are now only some small issues (mostly grammar mistakes) in the manuscript that still need to be addressed:

Lines 62-63: Please ensure tense consistency (either: were asked – they showed, or: are asked – they show).

Line 287: If I understand the sentence correctly, “significantly high” should be replaced by “significantly higher”.

Line 332: Did you mean “as less painful”? Please check.

Lines 337-339: I’m having difficulties understanding the sentence “Post hoc t-tests…”. Please check and rephrase for clarity.

Line 386: Replace “despite our findings” with “contrary to our findings” or “in contrast to our findings”.

Line 391: I think “either” is not needed here.

Lines 397-399: The part “or a masking or psychological feature as psychopathy boasts deceptive and manipulative capabilities” is unclear to me. Please revise.

Lines 401-402: Consider rephrasing the sentence as follows “Future research should examine this link more closely.” Also, in the preceding sentence, replace “ones’” with “one’s”, and “in psychopathy” with “in people with high psychopathic traits”.

Lines 412-413: Consider omitting “entirely”.

Lines 413-414: Consider rephrasing the sentence as follows: ”Although our finding was non-significant, the link between alexithymia and psychopathic traits should be explored further, as previous research found a strong support for this link (e.g. 4)”.

Lines 444-445: Please check the part “a accept higher levels…” and revise as appropriate.

Line 471: Consider rephasing as follows: “Lastly, although appropriate statistical tests were used…”.

Reviewer #4: I am sorry to have to say that most of my previous major concerns have not been met in this R2 version of the manuscript:

1) Wrong interpretation of the pain rating results: The authors compare self-pain ratings (NRS) between the groups for pressure stimuli whose intensity was individually adjusted so that they induced the same pain intensity rating in all participants in the first part of the experiment. Given this methodological approach, the objective pressure (measured in volts), rather than the subjective pain ratings (measured with a NRS) should be the main outcome of interest for the self-pain condition. It is therefore positive that the objective pressure results are now featured as a result rather than a manipulation check in this revised version of the manuscript.

However, the revision is not thorough and comprehensible enough. For example, the individually adjusted stimulus intensities are an important feature of the study and should be mentioned already in the abstract, otherwise the results cannot be understood correctly. Also, the logical reasoning of the manuscript throughout Methods, Results and Discussion is still that the pain ratings (NRS) rather than the pressure intensities (volts) should show a significant effect. Overall it appears that the authors do not understand the logic of their own experimental design.

2) Wrong use of terminology: The authors still fail to correctly distinguish between pain, which is by definition a subjective experience, and the objective stimulus pressure administered to induce that pain. This problem continues to cause confusion throughout the manuscript, as shown in the following example:

Line 280: „Additionally, we tested whether people higher in psychopathy accepted greater levels of high-pressure nociceptive pain (measured in volts) compared to the low psychopathic group…with… pain intensity level (high pain) as the dependent variable“. This sentence is confusing and wrong because a) volts are not a unit of pain but of pressure, b) indeed, the correct dependent variable here is pressure, not pain, and c) the phrasing „accepted“ insinuates that pain tolerance was measured when in fact participants simply had to rate how intense their pain perception was with each stimulus. This misleading and confusing wording is used throughout the manuscript, e.g. in the results section (Line 326-329). An example for a more correct phrasing would be e.g. „To test whether people higher in psychopathy required objectively more intense pressure stimuli (measured in volts) to report the same subjective pain intensity as the low psychopathic group, we performed a 2-way mixed ANOVA…“.

3) A new concern in this R2 version is that the presentation and discussion of results regarding self-pain ratings is contradictory throughout the manuscript:

The abstract and the response to reviewers letter suggest that the revised data analysis using ANOVA instead of multiple t-tests did not find a significant between-group difference in self-pain ratings any more:

Line 33 Abstract: „People higher in psychopathy did not self-report feeling less physical nociceptive pain compared to people lower in psychopathy“.

Reponse to reviewers: „We conducted the recommended ANOVA and self-pain ratings did not survive this test, thus, this is no longer a conclusion of the study“.

In contrast, the results session does report a highly significant between-group difference in self-pain ratings:

Line 316: „...the high psychopathy group reported less pain (M = 33.96, SD = 1.70) compared to the low psychopathy group (M = 40.40, SE = 1.60), t (47) = 2.75, p <.001“.

The high significance of this effect also contradicts the R1 version of this manuscript, where the t-test obtained only p=.03 (Line 429 in R1 manuscript).

This significant result is referred to throughout the Discussion, e.g.

Line 480 Discussion: "Additionally, we found people in the high psychopathy group self-reported experiencing less intense physical pain compared to people in the low psychopathy group".

But then again no effect seems to have been found:

Line 549: „we predicted that individuals with psychopathy would have reduced empathy for others’ pain as well as a accept higher levels of pain themselves. Our findings show support for the former, but not the latter„.

While in the conclusion the significant effect reemerges:

Line 593 Conclusion: „...psychopathic traits were associated with a reduced perception of nociceptive pain“.

This presentation is completely contradictory and inacceptable.

7. PLOS authors have the option to publish the peer review history of their article (what does this mean?). If published, this will include your full peer review and any attached files.

Reviewer #3: No

Reviewer #4: No

---

## [Author Response · Author response to Decision Letter 2]

31 May 2024

Dear reviewers,

Thank you for taking the time to re-review our manuscript. We greatly appreciate you taking the time to provide feedback. We have acknowledged the mistakes we made in the previously submitted manuscript and have hopefully now addressed all the comments that have been made to the correct standard. Below, you will find each of the comments that we received and the action we have taken. 

Thank you again for reviewing our work.

Reviewer #3: It is clear that the authors made a considerable effort to incorporate reviewers‘ feedback. In my opinion, there are now only some small issues (mostly grammar mistakes) in the manuscript that still need to be addressed:

Lines 62-63: Please ensure tense consistency (either: were asked – they showed, or: are asked – they show).

 Response: Thank you for your comment. These changes have been made. 

Line 287: If I understand the sentence correctly, “significantly high” should be replaced by “significantly higher”.

 Response: Thank you for your comment. This change has been made.

Line 332: Did you mean “as less painful”? Please check.

 Response: Thank you for picking up this mistake – the change has been made. 

Lines 337-339: I’m having difficulties understanding the sentence “Post hoc t-tests…”. Please check and rephrase for clarity.

 Response: Thank you for your comment. This change has been made.

Line 386: Replace “despite our findings” with “contrary to our findings” or “in contrast to our findings”.

Response: Thank you for your comment, this change has been made. 

Line 391: I think “either” is not needed here.

Response: Thank you, this has been removed. 

Lines 397-399: The part “or a masking or psychological feature as psychopathy boasts deceptive and manipulative capabilities” is unclear to me. Please revise.

Response: Thank you, this has been revised.

Lines 401-402: Consider rephrasing the sentence as follows “Future research should examine this link more closely.” Also, in the preceding sentence, replace “ones’” with “one’s”, and “in psychopathy” with “in people with high psychopathic traits”.

Response: Thank you, these changes have been made.

Lines 412-413: Consider omitting “entirely”.

 Response: This change has been made.

Lines 413-414: Consider rephrasing the sentence as follows: ”Although our finding was non-significant, the link between alexithymia and psychopathic traits should be explored further, as previous research found a strong support for this link (e.g. 4)”.

 Response: Thank you, this change has been made. 

Lines 444-445: Please check the part “a accept higher levels…” and revise as appropriate.

 Response: Thank you, this has been revised.

Line 471: Consider rephasing as follows: “Lastly, although appropriate statistical tests were used…”.

 Response: Thank you, this change has been made.

Reviewer #4: I am sorry to have to say that most of my previous major concerns have not been met in this R2 version of the manuscript:

1) Wrong interpretation of the pain rating results: The authors compare self-pain ratings (NRS) between the groups for pressure stimuli whose intensity was individually adjusted so that they induced the same pain intensity rating in all participants in the first part of the experiment. Given this methodological approach, the objective pressure (measured in volts), rather than the subjective pain ratings (measured with a NRS) should be the main outcome of interest for the self-pain condition. It is therefore positive that the objective pressure results are now featured as a result rather than a manipulation check in this revised version of the manuscript.

Response: Thank you for your comment.

However, the revision is not thorough and comprehensible enough. For example, the individually adjusted stimulus intensities are an important feature of the study and should be mentioned already in the abstract, otherwise the results cannot be understood correctly. 

Response: Thank you for your comments. The fact that pain is individually adjusted for each participant has been included in the abstract.

Also, the logical reasoning of the manuscript throughout Methods, Results and Discussion is still that the pain ratings (NRS) rather than the pressure intensities (volts) should show a significant effect. Overall it appears that the authors do not understand the logic of their own experimental design.

 Response: Thank you for your observation. I think as a team, internally, we were used to using the word ‘pain’ to encompass both the pressure stimuli and the subjective pain ratings while conducting the experiment, and then during the writing process, we still used them interchangeably rather than separating the two distinct subjects. Thank you for pointing out the distinction, and this has now been rectified throughout the manuscript. 

2) Wrong use of terminology: The authors still fail to correctly distinguish between pain, which is by definition a subjective experience, and the objective stimulus pressure administered to induce that pain. This problem continues to cause confusion throughout the manuscript, as shown in the following example:

Line 280: „Additionally, we tested whether people higher in psychopathy accepted greater levels of high-pressure nociceptive pain (measured in volts) compared to the low psychopathic group…with… pain intensity level (high pain) as the dependent variable“. 

This sentence is confusing and wrong because a) volts are not a unit of pain but of pressure, b) indeed, the correct dependent variable here is pressure, not pain, and c) the phrasing „accepted“ insinuates that pain tolerance was measured when in fact participants simply had to rate how intense their pain perception was with each stimulus. This misleading and confusing wording is used throughout the manuscript, e.g. in the results section (Line 326-329). An example for a more correct phrasing would be e.g. „To test whether people higher in psychopathy required objectively more intense pressure stimuli (measured in volts) to report the same subjective pain intensity as the low psychopathic group, we performed a 2-way mixed ANOVA…“.

Response: Thank you for your comment. After reading your example, we understand where we went wrong. We have amended this throughout the manuscript.

3) A new concern in this R2 version is that the presentation and discussion of results regarding self-pain ratings is contradictory throughout the manuscript:

The abstract and the response to reviewers letter suggest that the revised data analysis using ANOVA instead of multiple t-tests did not find a significant between-group difference in self-pain ratings any more:

Line 33 Abstract: „People higher in psychopathy did not self-report feeling less physical nociceptive pain compared to people lower in psychopathy“.

Response: Thank you for this observation, this mistake has been altered.

Reponse to reviewers: „We conducted the recommended ANOVA and self-pain ratings did not survive this test, thus, this is no longer a conclusion of the study“.

In contrast, the results session does report a highly significant between-group difference in self-pain ratings:

Line 316: „...the high psychopathy group reported less pain (M = 33.96, SD = 1.70) compared to the low psychopathy group (M = 40.40, SE = 1.60), t (47) = 2.75, p <.001“.

The high significance of this effect also contradicts the R1 version of this manuscript, where the t-test obtained only p=.03 (Line 429 in R1 manuscript).

Response: Thank you for your comment. Upon inspection, we made an error and an extra 0 was added to <.001 and instead it should be <.01, which has now been adjusted. 

This significant result is referred to throughout the Discussion, e.g.

Line 480 Discussion: "Additionally, we found people in the high psychopathy group self-reported experiencing less intense physical pain compared to people in the low psychopathy group".

But then again no effect seems to have been found:

Line 549: „we predicted that individuals with psychopathy would have reduced empathy for others’ pain as well as a accept higher levels of pain themselves. Our findings show support for the former, but not the latter„.

Response: Thank you for this observation, this has now been altered to show support for both predictions based on our significant finding.

While in the conclusion the significant effect reemerges:

Line 593 Conclusion: „...psychopathic traits were associated with a reduced perception of nociceptive pain“.

This presentation is completely contradictory and inacceptable.

Response: Thank you for being so observant, and we apologise for the confusion on our behalf and the error we made. We did in fact find a significant result and this has been consistently altered throughout the manuscript. We again apologise for our mistakes and thank you for taking the time to point out our errors.

---

## [Decision Letter · Decision Letter 3]

9 Jun 2024

PONE-D-23-10103R3Psychopathy, pain, and pain empathy: A psychophysiological study.PLOS ONE

Dear Dr. Alshukri,

Thank you for submitting your manuscript to PLOS ONE. After careful consideration, we feel that it has merit but does not fully meet PLOS ONE’s publication criteria as it currently stands. Therefore, we invite you to submit a revised version of the manuscript that addresses the points raised during the review process.

We look forward to receiving your revised manuscript.

Kind regards,

Vincenzo De Luca

Academic Editor

PLOS ONE

Journal Requirements:

Reviewers' comments:

Reviewer's Responses to Questions

**Comments to the Author**

1. If the authors have adequately addressed your comments raised in a previous round of review and you feel that this manuscript is now acceptable for publication, you may indicate that here to bypass the “Comments to the Author” section, enter your conflict of interest statement in the “Confidential to Editor” section, and submit your "Accept" recommendation.

Reviewer #3: (No Response)

2. Is the manuscript technically sound, and do the data support the conclusions?

Reviewer #3: Yes

3. Has the statistical analysis been performed appropriately and rigorously? 

Reviewer #3: Yes

4. Have the authors made all data underlying the findings in their manuscript fully available?

Reviewer #3: Yes

5. Is the manuscript presented in an intelligible fashion and written in standard English?

Reviewer #3: Yes

6. Review Comments to the Author

Reviewer #3: The authors have adequately addressed the issues found in the previous version. Overall, the revised manuscript reads well and is much clearer. I have identified only a small number of minor issues. I suggest authors resolving these issues to further improve the readability and clarity of the manuscript.

Abstract

Remove „growing“ in the penultimate sentence. Alternatively, you could write „We also show support for the theory of dual harm, which has been receiving increasing attention.“

In the last sentence, write either „consequently“ or „thus“. Using both is redundant.

Introduction

Page 4, line 85: Add a verb, e.g. “Fourth, mixed findings can be found in the current literature on psychopathy…” or “Fourth, current literature suggests mixed findings…”.

Page 5, line 124: Consider rephrasing as follows “…with little research looking into (or investigating) psychological aspects.” to avoid word repetition.

Discussion

Page 22, line 513-515: Consider rephrasing as follows “…suggest conducting an a priori power analysis and testing psychopathy groups within larger samples to increase statistical power.”

Page 22, line 527: For better clarity, replace “these traits” with “psychopathological traits”.

Page 22, line 532ff: Rephrase as follows “In conclusion, our study provides support for the dual harm model by demonstrating a diminished perception of pain in people higher in psychopathy. Additionally, we contribute to the current literature by showing that high psychopathic traits relate to problems in empathising with others’ pain.”

7. PLOS authors have the option to publish the peer review history of their article (what does this mean?). If published, this will include your full peer review and any attached files.

Reviewer #3: No

---

## [Author Response · Author response to Decision Letter 3]

11 Jun 2024

Dear Reviewer #3,

Thank you for again taking the time to review our document. We have made the changes you have recommended, and you will find each of the comments that we received and the action we have taken.

Thank you again for reviewing our work.

Reviewer #3: The authors have adequately addressed the issues found in the previous version. Overall, the revised manuscript reads well and is much clearer. I have identified only a small number of minor issues. I suggest authors resolving these issues to further improve the readability and clarity of the manuscript.

Abstract

Remove „growing“ in the penultimate sentence. Alternatively, you could write „We also show support for the theory of dual harm, which has been receiving increasing attention.“

Response: Thank you, this has been amended. 

In the last sentence, write either „consequently“ or „thus“. Using both is redundant.

Response: Thank you, this has been removed. 

Introduction

Page 4, line 85: Add a verb, e.g. “Fourth, mixed findings can be found in the current literature on psychopathy…” or “Fourth, current literature suggests mixed findings…”.

Response: Thank you, this change has been made. 

Page 5, line 124: Consider rephrasing as follows “…with little research looking into (or investigating) psychological aspects.” to avoid word repetition.

Response: Thank you, this change has been made. 

Discussion

Page 22, line 513-515: Consider rephrasing as follows “…suggest conducting an a priori power analysis and testing psychopathy groups within larger samples to increase statistical power.”

Response; Thank you, this change has been made. 

Page 22, line 527: For better clarity, replace “these traits” with “psychopathological traits”.

Response: Thank you, this change has been made. 

Page 22, line 532ff: Rephrase as follows “In conclusion, our study provides support for the dual harm model by demonstrating a diminished perception of pain in people higher in psychopathy. Additionally, we contribute to the current literature by showing that high psychopathic traits relate to problems in empathising with others’ pain.”

Response: Thank you, this change has been made.

---

## [Decision Letter · Decision Letter 4]

19 Jun 2024

Psychopathy, pain, and pain empathy: A psychophysiological study.

PONE-D-23-10103R4

Dear Dr. Alshukri,

We’re pleased to inform you that your manuscript has been judged scientifically suitable for publication and will be formally accepted for publication once it meets all outstanding technical requirements.

Kind regards,

Vincenzo De Luca

Academic Editor

PLOS ONE

Additional Editor Comments (optional):

Reviewers' comments:

Reviewer's Responses to Questions

**Comments to the Author**

1. If the authors have adequately addressed your comments raised in a previous round of review and you feel that this manuscript is now acceptable for publication, you may indicate that here to bypass the “Comments to the Author” section, enter your conflict of interest statement in the “Confidential to Editor” section, and submit your "Accept" recommendation.

Reviewer #3: All comments have been addressed

2. Is the manuscript technically sound, and do the data support the conclusions?

Reviewer #3: Yes

3. Has the statistical analysis been performed appropriately and rigorously? 

Reviewer #3: Yes

4. Have the authors made all data underlying the findings in their manuscript fully available?

Reviewer #3: Yes

5. Is the manuscript presented in an intelligible fashion and written in standard English?

Reviewer #3: Yes

6. Review Comments to the Author

Reviewer #3: (No Response)

7. PLOS authors have the option to publish the peer review history of their article (what does this mean?). If published, this will include your full peer review and any attached files.

Reviewer #3: No

---

## [Editor Report · Acceptance letter]

25 Jun 2024

PONE-D-23-10103R4 

PLOS ONE

Dear Dr. Alshukri, 

I'm pleased to inform you that your manuscript has been deemed suitable for publication in PLOS ONE. Congratulations! Your manuscript is now being handed over to our production team.

Kind regards, 

on behalf of

Dr. Vincenzo De Luca 

Academic Editor

PLOS ONE